# Genetic Effects of *LPIN1* Polymorphisms on Milk Production Traits in Dairy Cattle

**DOI:** 10.3390/genes10040265

**Published:** 2019-04-02

**Authors:** Bo Han, Yuwei Yuan, Ruobing Liang, Yanhua Li, Lin Liu, Dongxiao Sun

**Affiliations:** 1Department of Animal Genetics, Breeding and Reproduction, College of Animal Science and Technology, Key Laboratory of Animal Genetics, Breeding and Reproduction of Ministry of Agriculture and Rural Affairs, National Engineering Laboratory for Animal Breeding, China Agricultural University, Beijing 100193, China; bohan@cau.edu.cn (B.H.); workyyw0422@163.com (Y.Y.); liangruobing1992@163.com (R.L.); 2Beijing Dairy Cattle Center, Qinghe’nanzhen Deshengmenwai Street, Chaoyang District, Beijing 100192, China; 18800051836@163.com (Y.L.); liulin@bdcc.com.cn (L.L.)

**Keywords:** dairy cows, milk yield and composition, SNP, genetic association

## Abstract

Our initial RNA sequencing work identified that lipin 1 (*LPIN1*) was differentially expressed during dry period, early lactation, and peak of lactation in dairy cows, and it was enriched into the fat metabolic Gene Ontology (GO) terms and pathways, thus we considered *LPIN1* as the candidate gene for milk production traits. In this study, we detected the polymorphisms of *LPIN1* and verified their genetic effects on milk yield and composition in a Chinese Holstein cow population. We found seven SNPs by re-sequencing the entire coding region and partial flanking region of *LPIN1*, including one in 5′ flanking region, four in exons, and two in 3′ flanking region. Of these, four SNPs, c.637T > C, c.708A > G, c.1521C > T, and c.1555A > C, in the exons were predicted to result in the amino acid replacements. With the Haploview 4.2, we found that seven SNPs in *LPIN1* formed two haplotype blocks (D′ = 0.98–1.00). Single-SNP association analyses showed that SNPs were significantly associated with milk yield, fat yield, fat percentage, or protein yield in the first or second lactation (*p* = < 0.0001–0.0457), and only g.86049389C > T was strongly associated with protein percentage in both lactations (*p* = 0.0144 and 0.0237). The haplotype-based association analyses showed that the two haplotype blocks were significantly associated with milk yield, fat yield, protein yield, or protein percentage (*p* = < 0.0001–0.0383). By quantitative real-time PCR (qRT-PCR), we found that *LPIN1* had relatively high expression in mammary gland and liver tissues. Furthermore, we predicted three SNPs, c.637T > C, c.708A > G, and c.1521C > T, using SOPMA software, changing the LPIN1 protein structure that might be potential functional mutations. In summary, we demonstrated the significant genetic effects of *LPIN1* on milk production traits, and the identified SNPs could serve as genetic markers for dairy breeding.

## 1. Introduction

Genetic improvement can gradually induce smaller but permanent trait modifications [1], and genetic analyses have shown heritable variations in milk, fat and protein yields, and fat and protein percentages of bovine [2]. Genome-wide studies in dairy cows have allowed researchers to detect quantitative trait locus (QTL) harboring candidate polymorphic genes that are involved in milk production [3,4,5]. Moreover, studies have shown that the single nucleotide polymorphisms (SNPs) in the genes clearly influenced the milk yield and composition in Holsteins [6,7,8,9,10]. These results collectively suggest that loci found to affect milk production traits could be used in marker-assisted selection programs aimed at accelerating genetic progress of dairy cattle.

RNA sequencing (RNA-Seq) is an effective strategy to identify major genes for complex traits in human, domestic animals and plants, and previously we used it to analyze the liver transcriptome of Chinese Holstein cows among dry period, early lactation, and peak of lactation and found the expression of lipin 1 (*LPIN1*) significantly increased during peak of lactation compared to dry period (*p* = 0.00005, fold change = 2.02152) and during peak of lactation compared to early lactation (*p* = 0.00005, fold change = 1.30393). *LPIN1* was also enriched into the fat metabolic GO terms and pathways, including lipid and fatty acid metabolic processes, triglyceride mobilization, glycerolipid and glycerophospholipid metabolism, and mTOR signaling pathway [11]. LPIN1, a member of the lipin family proteins that paly critical roles in lipid metabolism, maintains the lipid metabolic homeostasis due to its dual functions on lipid metabolism. LPIN1 is a phosphatidate phosphatase enzyme that is required for lipid synthesis, and it can also act as a transcriptional co-activator to active fatty acid oxidation [12]. Studies have shown that the *LPIN1* links mTORC1 activity to the regulation of SREBP (sterol regulatory element-binding proteins) gene transcription [13,14,15], and that SREBPs integrate signals from multiple pathways to regulate fatty acids, triglycerides, and cholesterol synthesis [16,17]. *LPIN1* acts with PPARα (peroxisome proliferator-activated receptor α) and PGC-1α (PPARγ coactivator 1α) to modulate the hepatic genes involved in fatty acid oxidation [18]. PPARα, enriched in liver, is involved in hepatic glucose production, lipogenesis, and fatty acid transport and catabolism [19]. In addition, *LPIN1* is located 1.02 cM to the peak of the reported QTL region for milk fat yield [20], and is close to (0.0024–2.41 Mb) seven SNPs that have significant genetic effects on milk yield, fat yield, protein yield, fat percentage and protein percentage [21]. These data suggest that *LPIN1* might be involved in milk synthesis. Herein, we present the polymorphisms of *LPIN1* and their genetic effects on milk production traits in a Chinese Holstein cow population.

## 2. Materials and Methods

### 2.1. Animal Selection and Phenotypic Data Collection

We selected 1067 Chinese Holstein cows of 40 sire families from 22 dairy farms belonging to the Sanyuan Lvhe Dairy Farming Centre (Beijing, China) where the cows were maintained with the same feeding conditions. The sires were used for SNP discovery by PCR and sequencing and their 1067 daughters were for association analysis. The cows were selected according to the criteria as follows: (i) The cows in each sire family were distributed in different farms. (ii) At least six daughters were included in each sire family. (iii) Each cow had at least three generations of pedigree information and Dairy Herd Improvement (DHI) records for 305-day milk yield, protein yield, protein percentage, fat yield and fat percentage. The phenotypic data for the five traits during the first (1067 cows) and second (740 cows) lactations were provided by the Beijing Dairy Cattle Centre (http://www.bdcc.com.cn/; Beijing, China). The descriptive statistics of the phenotypic values for milk production traits in the two lactations are shown in Appendix A.

### 2.2. Genomic DNA Extraction

Genomic DNAs were extracted from the semen samples of 40 sires using the salt-out procedure, and the DNAs from the blood samples of 1067 daughters were extracted using a TIANamp Blood DNA Kit (Tiangen, Beijing, China) according to the manufacturer’s instructions. The quantity and quality of the extracted DNA samples were measured by a NanoDrop 2000 spectrophotometer (Thermo Scientific, Hudson, NH, USA) and gel electrophoresis, respectively.

### 2.3. SNP Identification and Estimation of Linkage Disequilibrium (LD)

We designed the primers by Primer3 (http://bioinfo.ut.ee/primer3-0.4.0/) to amplify the entire coding region and 2000 bp of the both 5′ and 3′ flanking regions based on the bovine reference genome sequences of *LPIN1* (Appendix A). The primers then were synthesized by the Beijing Genomics Institute (BGI, Beijing, China). The conditions for PCR are shown in Appendix A. Subsequently, the purified PCR products were sequenced using an ABI3730XL DNA analyzer (Applied Biosystems, Foster City, CA, USA). Then, we compared the sequences with the references on NCBI-BLAST (https://blast.ncbi.nlm.nih.gov/Blast.cgi) to discover the potential SNPs. The identified SNPs were individually genotyped using Sequenom MassArray for all the 1067 cows by matrix-assisted laser desorption/ionization time of flight mass spectrometry (MALDI-TOF MS, Sequenom MassARRAY, Bioyong Technologies Inc., HK).

Further, we used the Haploview 4.2 (Broad Institute of MIT and Harvard, Cambridge, MA, USA) to estimate the extent of Linkage Disequilibrium (LD) between the identified SNPs.

### 2.4. Association Analyses on Milk Production Traits

We utilized the SAS 9.13 software to estimate the associations of the SNPs or haplotype blocks with milk production traits on first or second lactation with the following animal model:(1)Y=μ+hys+b×M+G+a+e
where Y is the phenotypic value of each trait for each cow; μ is the overall mean; hys is the fixed effect of farm (1–22: 22 farms), year (1–4: 2012–2015), and season (1, April–May; 2, June–August; 3, September–November; and 4, December–March); b is the regression coefficient of covariant M; M is the fixed effect of calving month; G is the genotype or haplotype combination effect; a is the individual random additive genetic effect, distributed as $N\;\left(0,{A\delta }_{a}^{2}\right)$ with the additive genetic variance $\delta_{a}^{2}$; and e is the random residual, distributed as $N\;(0,\;{I\delta }_{e}^{2}$), with identity matrix I and residual error variance $\delta_{e}^{2}$. Bonferroni correction was performed for multiple testing, and the significant level was equal to the raw *P* value divided by number of genotypes or haplotype combinations.

We also calculated the additive (a), dominant (d), and substitution (α) effects as follows: $a=\frac{\mathrm{AA}-\mathrm{BB}}{2};\;d=\mathrm{AB}-\frac{\mathrm{AA}+\mathrm{BB}}{2};\;\alpha =a+d\left(q-p\right)$, where, AA, BB, and AB are the least square means of the milk production traits in the corresponding genotypes, p is the frequency of allele A, and q is the frequency of allele B.

### 2.5. Detection of LPIN1 Gene Expression by Quantitative Real-Time PCR (qRT-PCR)

To further reveal the potential function of *LPIN1*, we detected its expressions in eight tissues, including heart, liver, spleen, lung, kidney, ovary, mammary, and uterus. We selected three healthy lactating Chinese Holstein cows from the Sanyuanlvhe Dairy Farming Center (Beijing, China), and collected the eight tissues from each cow. The total RNAs from the eight tissues of each cow were isolated using Trizol reagent (Invitrogen, Carlsbad, CA, USA) according to the manufacturer’s instructions. Subsequently, we measured the quantity and quality of RNA with the NanoDrop 2000 spectrophotometer (Thermo Scientific, Hudson, DE, USA) and gel electrophoresis, respectively. We used PrimerScriptH RT reagent Kit (TaKaRa Biotechnology Co., Ltd., Dalian, China) for the reverse transcription. The qRT-PCR primers of *LPIN1* and two reference genes (Ribosomal Protein S9, *RPS9* and Ubiquitously Expressed Prefoldin Like Chaperone, *UXT*) are shown in Appendix A. Then, we conducted the qRT-PCR in a LightCycler^®^ 480 II (Roche, Penzberg, Germany) using SYBR green fluorescence (Roche) as the procedures presented in Appendix A. We performed all the measurements in triplicate and the relative gene expression was normalized by the *RPS9* and *UXT* with 2^−ΔΔCt^ method.

### 2.6. Protein Structure Prediction

The NPSA SOPMA Server (https://npsa-prabi.ibcp.fr/cgi-bin/npsa_automat.pl?page=/NPSA/npsa_sopma.html) was used to predict the variation of protein secondary structure caused by the missense mutation in the coding region of *LPIN1* gene, and the parameters were window width (17), similarity threshold (8), and number of states (4).

### 2.7. Ethics Approval

All protocols for sample collection were approved by the Institutional Animal Care and Use Committee (IACUC) at China Agricultural University (Beijing, China; permit number: DK996).

## 3. Results

### 3.1. SNPs Identification in LPIN1

By sequencing and sequence alignment, we identified seven SNPs in *LPIN1* gene: one SNP in 5′ flanking region, four in exons, and two in 3′ flanking region. The genotypic and allelic frequencies of all the identified SNPs are summarized in Table 1. In addition, the four SNPs in the exons were predicted to result in amino acid replacements, namely, c.637T > C (ATG–ACG; methionine to threonine), c.708A > G (ACA–GCA; threonine to alanine), c.1521C > T (CCC–TCC; proline to serine), and c.1555A > C (CAT–CCT; histidine to proline; Table 1).

### 3.2. Association Analysis Between SNPs and the Five Milk Production Traits

Associations between seven SNPs of *LPIN1* and five milk production traits are presented in Table 2. Using the single-SNP association analysis, we found that two (g.86049523C > T and g.86049389C > T), four (g.86129263C > G, c.1521C > T, g.86049523C > T and g.86049389C > T), four (g.86129263C > G, c.1521C > T, g.86049523C > T and g.86049389C > T), and one (g.86049389C > T) SNPs were significantly associated with milk yield (*p* < 0.0001), fat yield (*p* = < 0.0001–0.0298), protein yield (*p* = < 0.0001–0.0326), and protein percentage (*p* = 0.0237) in the first lactation, respectively. Similarly, five (c.637T > C, c.708A > G, c.1521C > T, c.1555A > C, and g.86049523C > T), two (g.86049523C > T and g.86049389C > T), four (g.86129263C > G, c.637T > C, c.1521C > T, and c.1555A > C), four (c.637T > C, c.708A > G, c.1555A > C, and g.86049523C > T), and one (g.86049523C > T) SNPs were significantly associated with milk yield (*p* = < 0.0001–0.0239), fat yield (*p* = 0.0053 and 0.0236), fat percentage (*p* = 0.0087–0.0457), protein yield (*p* = 0.0047–0.0281), and protein percentage (*p* = 0.0144) in the second lactation, respectively. The SNP g.86049523C > T was identified significantly associated with milk yield, fat yield, and protein yield in both first and second lactations (*p* = < 0.0001–0.0281), and g.86049389C > T was strongly associated with fat yield in both lactations (*p* = 0.0053–0.0023). Overall, seven SNPs of *LPIN1* (g.86129263C > G, c.637T > C, c.708A > G, c.1521C > T, c.1555A > C, g.86049523C > T, and g.86049389C > T) were observed strongly associated with at least one milk production traits in the first and second lactations. Nevertheless, one (g.86129263C > G), three (c.637T > C, c.708A > G, and c.1555A > C), three (c.708A > G, g.86049523C > T, and g.86049389C > T), and five SNPs (g.86129263C > G, c.637T > C, c.708A > G, c.1521C > T, and c.1555A > C) had no genetic associations with milk yield, fat yield, fat percentage, and protein percentage in either lactation, respectively (*p* > 0.05).

In addition, we analyzed the additive, dominant and substitution effects of the seven SNPs in the first and second lactations, and found that the SNPs were significantly associated with at least one milk traits in the first or second lactation (*p* < 0.05; Appendix A).

### 3.3. Associations Between Haplotype Blocks and the Five Milk Traits

We used Haploview 4.2 to estimate the extent of LD between the seven identified SNPs, and inferred two haplotype blocks, respectively, comprising two and five SNPs (Figure 1). The pairwise D′ measures indicated that SNPs in the block were highly linked (Block 1: D′ = 1; Block 2: D′ = 0.98–1.00). In Block 1, the frequency of the three haplotypes, H1 (CC), H2 (TT), and H3 (TC), were 76.1%, 10.55%, and 13.4%, respectively. For Block 2, three haplotypes, H1 (ACATC), H2 (CTGCG), and H3 (CCGCC), with frequencies of 41.9%, 27.4%, and 29.1%, respectively, were identified. Haplotype-based association analysis showed that the haplotype Block 1 was significantly associated milk yield (*p* < 0.0001), fat yield (*p* = < 0.0001), and protein yield (*p* = < 0.0001) in the first lactations, and associated with fat yield (*p* = 0.0185) and protein percentage (*p* = 0.01) in the second lactation. The haplotype Block 2 was significantly associated with milk yield (*p* = 0.002 and 0.0044) and protein yield (*p* = 0.0383 and 0.0052) in both lactations (Table 3). While haplotype Block 1 had no significant association with fat percentage, Block 2 had no association with fat yield, fat percentage, and protein percentage in both lactations (*p* > 0.05).

### 3.4. Expression of LPIN1 in Eight Tissues

Using qRT-PCR, we detected that *LPIN1* gene was expressed in heart, liver, spleen, lung, kidney, ovary, mammary, and uterus of lactating Chinese Holstein, with relatively high expression in the mammary gland and liver, implying the *LPIN1* might be involved in milk synthesis (Figure 2).

### 3.5. Protein Structure Variation Caused by the Missense Mutation

We further used the SOPMA software to predict the changes of the protein secondary structure of LPIN1 for the four missense mutations in the exons of *LPIN1* gene, and found three missense mutations (c.637T > C, c.708A > G, and c.1521C > T) that changed the LPIN1 protein secondary structure (Table 1). Of these, the missense mutation (methionine to threonine) caused by c.637T > C led to the alpha helix changing 25.7–25.81%, extended strand 18.44–17.88%, and random coil 48.83–49.27%. The c.708A > G caused a missense mutation (threonine to alanine) to result in the changes of alpha helix of 25.7–26.03%, extended strand of 18.44–18.32%, and random coil of 48.83–48.6%. Due to the missense mutation proline to serine, caused by c.1521C > T, the alpha helix changed 25.7–25.81% and random coil 48.83–48.72%. However, the missense mutation caused by c.1555A > C, histidine to proline, did not change the protein secondary structure of LPIN1.

## 4. Discussion

Our initial RNA-Seq study suggested *LPIN1* as the promising candidate gene for milk production traits in dairy cattle, and this follow-up investigation demonstrated the significant genetic associations of *LPIN1* with milk, fat and protein yields.

Studies showing the important roles of SNPs for milk production traits in dairy cattle have been reported previously [8,9,10,22]. Pegolo et al. discovered two SNPs, rs136905033 and rs137457402, of *LPIN1* associated with milk fatty acids in bovine [23]. Nafikov et al. did not observe any significant association between *LPIN1* haplotypes and the milk fatty acid composition in Holstein cows [24]. In this study, we identified the polymorphisms of *LPIN1* and found that the SNPs and haplotype blocks of this gene have strongly associations with milk production traits, especially milk, fat and protein yields. Further, based on the phenotypic values of the genotypes for each SNP (Table 2), we found that the genotype TT of c.1521C > T with higher milk production had a positive effect on milk yield. For milk fat, the genotypes GG of g.86129263C > G and CC of g.86049389C > T were positively influenced the fat yield, and the genotypes TT of c.637T > C and AA of c.1555A > C have more active effects on fat percentage. As for milk protein, we observed that the cows with genotypes GG of g.86129263C > G and TT of c.1521C > T produced more milk protein. In summary, our findings provide information for improving the milk production in dairy cattle.

In the current study, we identified *LPIN1* as highly expressed in the mammary gland and liver tissues. The lactating bovine mammary gland is regarded as a formidable machine for triacylglycerol synthesizing. LPIN1 is the major isoform expressed in the bovine mammary, and it catalyzes phosphatidic acid to diacylglycerol in triacylglycerol synthesis [25,26]. *LPIN1*, a transcriptional co-activator for regulating fatty acid metabolism, influences lipid metabolism and glucose homeostasis in various tissues including liver, adipose tissue and some other cells [27]. The mRNA expression of *LPIN1* in liver and adipose tissue have positive associations with body mass and insulin sensitivity, and its polymorphisms have been associated with metabolic syndrome [28]. The allele T of rs13412852 in *LPIN1* is associated with lower body mass index and insulin levels [29], thus it is possibly considered as a protector for the metabolic syndrome alterations. These data suggest that *LPIN1* might be involved in milk synthesis and in regulating the transcription of other genes involved in milk synthesis metabolism.

Furthermore, we found four missense mutations in the exons of *LPIN1*, and the association analysis results showed that the four mutations were significantly associated with milk yield, fat yield, fat percentage, or protein yield. Importantly, we predicted the protein secondary structure changes by these four mutations, and found that three missense mutations, c.637T > C, c.708A > G, and c.1521C > T, could change the LPIN1 protein structure by altering the alpha helix, extended strand, and random coil. Generally, the α-helix was preferably located at the core of the protein and had important roles in proteins for flexibility and conformational changes [30]. Therefore, we presumed that the LPIN1 protein might be more stable in conformation when the haplotype was H2 in Block 2, with the base C in c.637T > C, G in c.708A > G, and T in c.1521C > T. Hence, these three missense mutations might be the potential functional mutations for milk production by changing the protein structure in dairy cattle, and the further in-depth investigation should be performed to validate their biological roles.

## 5. Conclusions

In this study, we identified seven SNPs in *LPIN1* and observed their significant genetic associations with milk production traits in dairy cattle. Three missense mutations, c.637T > C, c.708A > G, and c.1521C > T, might be functional mutations by changing the LPIN1 protein structure. Our findings provide worthwhile information that could be used in breeding programs.

## Figures and Tables

**Figure 1 genes-10-00265-f001:**
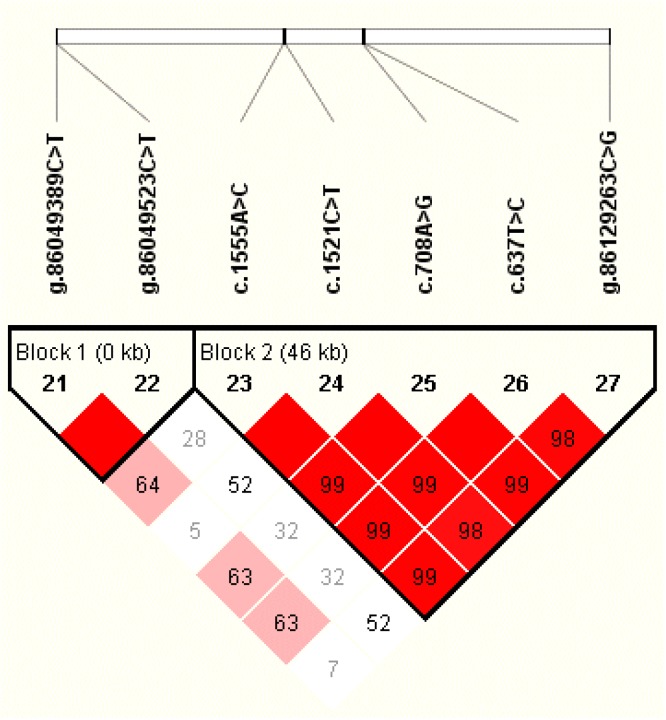
Linkage disequilibrium estimated among the SNPs in *LPIN1* (D′ = 0.98–1.00). Seven SNPs of *LPIN1* formed two haplotype blocks. The values in boxes are pairwise SNP correlations (D′), while bright red boxes without numbers indicate complete LD (D′ = 1). The blocks indicate haplotype blocks and the text above the horizontal numbers is the SNP names.

**Figure 2 genes-10-00265-f002:**
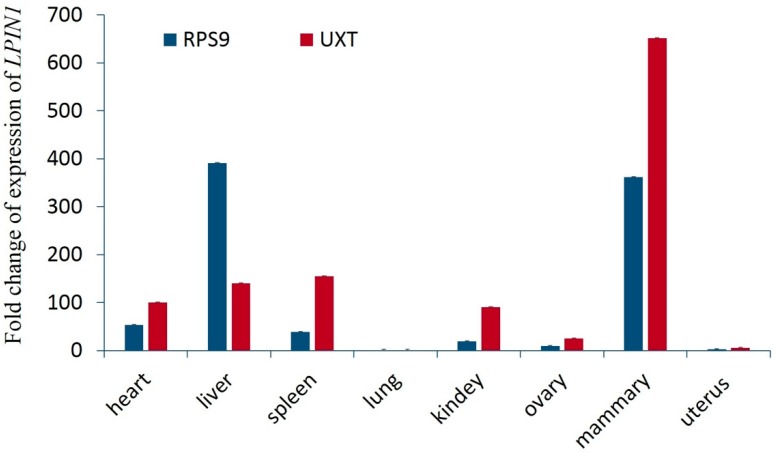
Expression profiles of *LPIN1* gene in eight different tissues. *RPS9* and *UXT* were the two reference genes.

**Table 1 genes-10-00265-t001:** Details of single nucleotide polymorphisms (SNPs) identified in *LPIN1* gene.

SNP name	Alias	Location	Position (UMD3.1)	GenBank no.	Genotype	Genotypic Frequency	Allele	Allelic Frequency	Codon	Amino Acid	Alpha Helix (%)	Extended Strand (%)	Beta Turn (%)	Random Coil (%)
g.86129263C > G		5′ flanking region	chr11:86129263	rs211527179	CC	0.52	C	0.72						
					CG	0.40	G	0.28						
					GG	0.08								
c.637T > C	p.Met103Thr	exon 5	chr11:86093896	rs110871255	TT	0.35	T	0.58	ATG	Methionine	25.7	18.44	7.04	48.83
					TC	0.46	C	0.42	ACG	Threonine	25.81	17.88	7.04	49.27
					CC	0.19								
c.708A > G	p.Thr127Ala	exon 5	chr11:86093825	rs110161110	AA	0.18	A	0.42	ACA	Threonine	25.7	18.44	7.04	48.83
					AG	0.47	G	0.58	GCA	Alanine	26.03	18.32	7.04	48.6
					GG	0.35								
c.1521C > T	p.Pro398Ser	exon 8	chr11:86082411	rs207681322	CC	0.52	C	0.72	CCC	Proline	25.7	18.44	7.04	48.83
					CT	0.40	T	0.28	TCC	Serine	25.81	18.44	7.04	48.72
					TT	0.08								
c.1555A > C	p.His409Pro	exon 8	chr11:86082377	rs137642654	AA	0.18	A	0.41	CAT	Histidine	25.7	18.44	7.04	48.83
					AC	0.46	C	0.59	CCT	Proline	25.7	18.44	7.04	48.83
					CC	0.35								
g.86049523C > T		3′ flanking region	chr11:86049523	rs135886289	CC	0.80	C	0.89						
					TC	0.19	T	0.11						
					TT	0.01								
g.86049389C > T		3′ flanking region	chr11:86049389	rs109039955	CC	0.57	C	0.76						
					CT	0.38	T	0.24						
					TT	0.05								

**Table 2 genes-10-00265-t002:** Associations of seven SNPs with milk production traits in two lactations in Chinese Holstein (LSM ± SE).

SNP	Lactation	Genotype (No.)	Milk Yield (kg)	Fat Yield (kg)	Fat Percentage (%)	Protein Yield (kg)	Protein Percentage (%)
g.86129263C > G	1	CC (549)	10,277 ± 64.99	339.78 ± 2.89 ^a^	3.33 ± 0.03	302.37 ± 2.1 ^a^	2.95 ± 0.01
		GC (418)	10,373 ± 66.2	343.97 ± 2.93 ^b^	3.33 ± 0.03	305.94 ± 2.13 ^b^	2.96 ± 0.01
		GG (84)	10,427 ± 102	348.19 ± 4.28 ^b^	3.36 ± 0.04	308.25 ± 3.12 ^b^	2.96 ± 0.01
		*p*	0.0897	0.0264 *	0.7395	0.0171 *	0.6191
	2	CC (393)	10,830 ± 66.31	394.13 ± 2.94	3.65 ± 0.03 ^a^	322.86 ± 2.14	2.98 ± 0.01
		GC (279)	10,909 ± 70.55	389.92 ± 3.1	3.57 ± 0.03 ^b^	323.77 ± 2.26	2.96 ± 0.01
		GG (56)	11,027 ± 121.7	396.24 ± 5.05	3.58 ± 0.05 ^ab^	328.01 ± 3.68	2.97 ± 0.02
		*p*	0.1742	0.1778	0.0114 *	0.321	0.3621
c.637T > C	1	CC (355)	10,306 ± 67.75	341.19 ± 2.99	3.33 ± 0.03	304.44 ± 2.18	2.96 ± 0.01
		CT (471)	10,298 ± 65.56	342.24 ± 2.91	3.34 ± 0.03	303.55 ± 2.12	2.95 ± 0.01
		TT (195)	10,260 ± 79.47	340.9 ± 3.43	3.34 ± 0.03	301.61 ± 2.5	2.95 ± 0.01
		*p*	0.8001	0.8326	0.8354	0.3977	0.3462
	2	CC (245)	10,822 ± 74.61 ^ab^	387.99 ± 3.26	3.59 ± 0.03 ^a^	320.57 ± 2.38 ^ab^	2.96 ± 0.01
		CT (334)	10,915 ± 69.65 ^a^	393.46 ± 3.08	3.61 ± 0.03 ^ab^	324.8 ± 2.24 ^a^	2.98 ± 0.01
		TT (134)	10,695 ± 88.29 ^b^	391.37 ± 3.79	3.68 ± 0.04 ^b^	318.59 ± 2.76 ^b^	2.98 ± 0.01
		*p*	0.0239 *	0.1384	0.0457 *	0.0135 *	0.3878
c.708A > G	1	AA (189)	10,306 ± 79.73	341.1 ± 3.43	3.33 ± 0.03	302.25 ± 2.5	2.94 ± 0.01
		GA (483)	10,283 ± 65.18	341.71 ± 2.9	3.34 ± 0.03	302.53 ± 2.11	2.95 ± 0.01
		GG (363)	10,288 ± 67.68	339.86 ± 2.99	3.33 ± 0.03	302.91 ± 2.18	2.96 ± 0.01
		*p*	0.9415	0.6967	0.7041	0.9482	0.4343
	2	AA (131)	10,585 ± 87.91 ^a^	385.53 ± 3.75	3.67 ± 0.04	315.15 ± 2.73 ^a^	2.98 ± 0.01
		GA (345)	10,820 ± 67.54 ^b^	388.86 ± 2.99	3.61 ± 0.03	321.28 ± 2.17 ^b^	2.98 ± 0.01
		GG (249)	10,700 ± 74.6 ^ab^	382.4 ± 3.27	3.59 ± 0.03	315.81 ± 2.38 ^a^	2.96 ± 0.01
		*p*	0.0108 *	0.06	0.0578	0.0047 **	0.1921
c.1521C > T	1	CC (548)	10,353 ± 64.39	342.88 ± 2.86 ^a^	3.33 ± 0.03	305.29 ± 2.08 ^a^	2.95 ± 0.01
		CT (424)	10,416 ± 66.09	346.9 ± 2.93 ^ab^	3.34 ± 0.03	308.3 ± 2.13 ^b^	2.96 ± 0.01
		TT (82)	10,529 ± 102.6	351.26 ± 4.3 ^b^	3.35 ± 0.04	311.14 ± 3.13 ^b^	2.96 ± 0.01
		*P*	0.1277	0.0298 *	0.8969	0.0326 *	0.5631
	2	CC (391)	10,794 ± 66.73 ^Aa^	389.16 ± 2.92	3.66 ± 0.03 ^a^	320.29 ± 2.13	2.98 ± 0.01
		CT (286)	10,752 ± 71.54 ^Aa^	383.27 ± 3.13	3.59 ± 0.03 ^b^	319.05 ± 2.28	2.97 ± 0.01
		TT (56)	11,321 ± 125.69 ^B^	387.58 ± 5.11	3.56 ± 0.05 ^ab^	324.06 ± 3.73	2.95 ± 0.02
		*p*	<0.0001 **	0.0734	0.0087 **	0.343	0.4776
c.1555A > C	1	AA (190)	10,318 ± 79.45	342.83 ± 3.42	3.34 ± 0.03	303.86 ± 2.49	2.95 ± 0.01
		CA (484)	10,276 ± 64.88	341.58 ± 2.88	3.34 ± 0.03	303.41 ± 2.1	2.96 ± 0.01
		CC (367)	10,289 ± 68.17	341.3 ± 3.01	3.33 ± 0.03	304.43 ± 2.19	2.96 ± 0.01
		*p*	0.8129	0.8592	0.9428	0.8119	0.3199
	2	AA (129)	10,608 ± 88.23 ^Aa^	390.24 ± 3.77	3.7 ± 0.04 ^a^	316.59 ± 2.74 ^a^	2.99 ± 0.01
		CA (349)	10,875 ± 68.16 ^Bb^	393.78 ± 3.01	3.63 ± 0.03 ^ab^	322.98 ± 2.19 ^b^	2.98 ± 0.01
		CC (249)	10,772 ± 73.77 ^ab^	387.22 ± 3.22	3.6 ± 0.03 ^b^	318.36 ± 2.35 ^a^	2.96 ± 0.01
		*p*	0.005 **	0.0515	0.023 *	0.0087 **	0.1266
g.86049523C > T	1	CC (849)	10,240 ± 60.9 ^Aa^	338 ± 2.73 ^Aa^	3.32 ± 0.03	302.19 ± 1.99 ^Aa^	2.96 ± 0.01
		TC (200)	10,560 ± 77.78 ^Bb^	352.15 ± 3.36 ^Bb^	3.35 ± 0.03	311.83 ± 2.45 ^Bb^	2.96 ± 0.01
		TT (12)	10,457 ± 242.54 ^ab^	330.84 ± 9.83 ^ab^	3.21 ± 0.1	303.49 ± 7.17 ^ab^	2.92 ± 0.04
		*p*	<0.0001 **	<.0001 **	0.2535	<0.0001 **	0.4739
	2	CC (595)	10,802 ± 63.12 ^a^	388.88 ± 2.84 ^a^	3.61 ± 0.03	322.08 ± 2.05 ^a^	2.97 ± 0.01 ^a^
		TC (137)	10,647 ± 86.89 ^ab^	385.9 ± 3.71 ^ab^	3.65 ± 0.04	320.52 ± 2.73 ^a^	3 ± 0.01 ^Bb^
		TT (5)	9933.02 ± 349.64 ^b^	351.83 ± 14.12 ^b^	3.54 ± 0.14	294.92 ± 10.4 ^b^	2.98 ± 0.05 ^ab^
		*p*	0.0091 **	0.0236 *	0.4408	0.0281 *	0.0144 *
g.86049389C > T	1	CC (590)	10,320 ± 62.88 ^Aa^	343.46 ± 2.8 ^Aa^	3.34 ± 0.03	305.67 ± 2.04 ^Aa^	2.96 ± 0.01 ^a^
		CT (394)	10,562 ± 66.66 ^Bb^	350.18 ± 2.95 ^Bb^	3.32 ± 0.03	313.12 ± 2.15 ^Bb^	2.96 ± 0.01 ^a^
		TT (56)	10,518 ± 118.53 ^ab^	341.71 ± 4.9 ^ab^	3.27 ± 0.05	306.2 ± 3.57 ^ab^	2.92 ± 0.02 ^b^
		*p*	<0.0001 **	0.0023 **	0.2859	<0.0001 **	0.0237 *
	2	CC (416)	10,783 ± 65.86	391.43 ± 2.93 ^a^	3.64 ± 0.03	319.02 ± 2.14	2.97 ± 0.01
		CT (277)	10,725 ± 71.14	384.63 ± 3.13 ^b^	3.61 ± 0.03	318.2 ± 2.28	2.98 ± 0.01
		TT (35)	10,916 ± 149.18	399.1 ± 6.14 ^a^	3.68 ± 0.06	326.33 ± 4.48	3 ± 0.02
		*p*	0.3522	0.0053 **	0.2585	0.1662	0.1656

Note: The number in the bracket represents the number of cows for the corresponding genotype; *p* shows the significance for the genetic effects of SNPs; ^a,b^ within the same column with different superscripts means *p* < 0.05; ^A,B^ within the same column with different superscripts means *p* < 0.01; * indicates *p* < 0.05; ** indicates *p* < 0.01.

**Table 3 genes-10-00265-t003:** Associations of haplotype blocks with milk production traits in two lactations in Chinese Holstein (LSM ± SE).

Block	Lactation	Haplotype Combination (No.)	Milk Yield (kg)	Fat Yield (kg)	Fat Percentage (%)	Protein Yield (kg)	Protein Percentage (%)
*LPIN1*-1	1	H1H1 (614)	10,244 ± 63.83 ^Aa^	341.58 ± 2.85 ^Aa^	3.35 ± 0.03	302.9 ± 2.08 ^Aa^	2.96 ± 0.01
		H1H2 (174)	10,592 ± 80.33 ^Bb^	355.63 ± 3.45 ^B^	3.36 ± 0.03	314.22 ± 2.52 ^B^	2.96 ± 0.01
		H1H3 (223)	10,374 ± 76.44 ^a^	340.98 ± 3.32 ^Aa^	3.31 ± 0.03	306.43 ± 2.42 ^Aa^	2.96 ± 0.01
		*p*	<0.0001 **	<0.0001 **	0.1356	<0.0001 **	0.7979
	2	H1H1 (426)	10,820 ± 66.83	390.47 ± 2.98 ^a^	3.62 ± 0.03	320.89 ± 2.17	2.97 ± 0.01 ^a^
		H1H2 (121)	10,695 ± 92.24	384.18 ± 3.95 ^ab^	3.61 ± 0.04	320.77 ± 2.87	3 ± 0.01 ^b^
		H1H3 (158)	10,797 ± 84.63	382.69 ± 3.64 ^b^	3.56 ± 0.03	318.51 ± 2.65	2.96 ± 0.01 ^a^
		*p*	0.3532	0.0185 *	0.1696	0.5462	0.01 *
*LPIN1*-2	1	H1H1 (201)	10,195 ± 81.19 ^Aa^	339.42 ± 3.38	3.32 ± 0.03	303.29 ± 2.49 ^ac^	2.94 ± 0.01
		H1H2 (227)	10,498 ± 78.47 ^Bb^	346.15 ± 3.3	3.35 ± 0.03	307.55 ± 2.39 ^bc^	2.95 ± 0.01
		H1H3 (247)	10,299 ± 76.23 ^ab^	340.2 ± 3.19	3.33 ± 0.03	302.72 ± 2.35 ^a^	2.95 ± 0.01
		H2H2 (80)	10,398 ± 110.37 ^ab^	348.02 ± 4.38	3.36 ± 0.04	307.48 ± 3.2 ^abc^	2.96 ± 0.01
		H2H3 (183)	10,332 ± 80.05 ^ab^	342.19 ± 3.37	3.32 ± 0.03	308.35 ± 2.47 ^b^	2.96 ± 0.01
		H3H3 (93)	10,178 ± 104.38 ^a^	341.58 ± 4.09	3.36 ± 0.04	302.3 ± 3.04 ^ac^	2.95 ± 0.01
		*p*	0.002 **	0.1276	0.8047	0.0383 *	0.8859
	2	H1H1 (134)	10,632 ± 88.27 ^a^	387.72 ± 3.79	3.66 ± 0.04	315.83 ± 2.76 ^Aa^	2.98 ± 0.01
		H1H2 (154)	10,907 ± 84.15 ^ab^	390.43 ± 3.63	3.58 ± 0.03	323.03 ± 2.64 ^Cc^	2.97 ± 0.01
		H1H3 (184)	10,944 ± 80.55 ^b^	394.99 ± 3.49	3.62 ± 0.03	323.2 ± 2.54 ^BCbc^	2.96 ± 0.01
		H2H2 (54)	10,909 ± 123.9 ^ab^	390.82 ± 5.15	3.58 ± 0.05	322.46 ± 3.76 ^abc^	2.97 ± 0.02
		H2H3 (121)	10,748 ± 89.14 ^ab^	383.7 ± 3.81	3.57 ± 0.04	315.33 ± 2.78 ^Aa^	2.94 ± 0.01
		H3H3 (68)	10,700 ± 109.38 ^ab^	387.14 ± 4.57	3.61 ± 0.04	317.31 ± 3.33 ^abc^	2.97 ± 0.02
		*p*	0.0044 **	0.0708	0.2162	0.0052 **	0.3756

Note: H means haplotype; the number in the bracket represents the number of cows for the corresponding haplotype combination; *LPIN1*-1: H1(CC), H2(TT), H3(TC); *LPIN1*-2: H1(ACATC), H2(CTGCG), H3(CCGCC); *p* shows the significance for genetic effects among the haplotype blocks; * indicates *p* < 0.05; ** indicates *p* < 0.01; ^a,b,c^ within the same column with different superscripts means *p* < 0.05; ^A,B,C^ within the same column with different superscripts means *p* < 0.01.

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
