# Peer review of "Genetic Effects of LPIN1 Polymorphisms on Milk Production Traits in Dairy Cattle"

_genes, 2019, doi:10.3390/genes10040265_

Reviewer 1 Report

I have no more comments

Author Response

Thank you for your approval.

Reviewer 2 Report

This is a good extension of their earlier studies using LPIN1. Especially appreciate the use of haplotypes to assess the association of this gene with production traits.

Needs some minor work. For example, on Page 9, line 169, remove "the" from in front of Haploview.

Also,an exaggerated statement on line 178, "block 2 was remarkable associated" should be edited.

Author Response

Point 1: For example, on Page 9, line 169, remove "the" from in front of Haploview.

Response 1: As suggested, we have deleted “the” in front of Haploview.

Point 2: Also, an exaggerated statement on line 178, "block 2 was remarkable associated" should be edited.

Response 2: Thanks for your comment. We have revised the “remarkable” to “significantly” (please see page 9, line 186).

Reviewer 3 Report

Line 36: I suggest to support the thesis about permanent trait modification with a reference [Visentin, G., Niero, G., Berry, D. P., Costa, A., Cassandro, M., De Marchi, M., & Penasa, M. (2019). Genetic (co) variances between milk mineral concentration and chemical composition in lactating Holstein-Friesian dairy cows. animal13(3), 477-486].

Line 44: in the previous work the focus was on RNA. On the other hand, the present work focuses on DNA. For this reason you have to explain better why a new work based on DNA is needed on this stage.

Line 59 to line 62: sentences about existing data and the lack of studies are contraddictory.

Line 71: on average how many test-day per cow?

Line 72 and 73: you are referring to two lactations. Meaning that cows of first and second lactation were selected? How many in first and how many in second lactation? Clarify.

Line 94: full name of "LD" acronym is needed.

Line 98: is there a specific reason why you did not inserted days in milk and parity of animals in the model? I think these two fixed effects are very important.

Line 99 to line 103: specify the number of levels for each fixed effect inserted in the model.

Line 145: To me, it is not clear what you mean for associations. Further than the significance of the association (P value), you should state and discuss whenever the association is positive or negative for each one of the traits.

Line147: Not clear what you mean for "two, four, four, and one SNPs". Please clarify here and in the rest of the manuscript.

Line 191: Expression of LPIN1 is missing in the graph. Unit of measure of y-axis is missing.

Line 210: the discussion of results sounds more like an introduction. I suggest to discuss more your results in comparison with literature instead of being so general.

Line 211 to line 213: I don t think that your work is the first on the topic (See Pegolo et al. for example). 

Line 223: full name for "FA" acronym is needed.

Author Response

Point 1: Line 36: I suggest to support the thesis about permanent trait modification with a reference [Visentin, G., Niero, G., Berry, D. P., Costa, A., Cassandro, M., De Marchi, M., & Penasa, M. (2019). Genetic (co) variances between milk mineral concentration and chemical composition in lactating Holstein-Friesian dairy cows. animal, 13(3), 477-486].

Response 1: Thanks for your suggestion. We have added the reference (please see page 1, line 36, and page 17, lines 280-282).

Point 2: Line 44: in the previous work the focus was on RNA. On the other hand, the present work focuses on DNA. For this reason you have to explain better why a new work based on DNA is needed on this stage.

Response 2: Thanks for your comment. In the previous work, we used the RNA sequencing to identify the promising candidate genes for milk production traits, and we did not know whether they really affected the milk synthesis or not. In this study, we confirmed the genetic effect of LPIN1 polymorphisms on milk traits based on the association analyses.

 Point 3: Line 59 to line 62: sentences about existing data and the lack of studies are contradictory.

Response 3: Thanks for your comment. We have rephrased the sentences as follows: “These data suggest that LPIN1 might be involved in milk synthesis. Herein, we will detect the polymorphisms of LPIN1 and confirm their genetic effects on milk production traits in a Chinese Holstein cow population.” (Please see page 3, lines 64-66)

 Point 4: Line 71: on average how many test-day per cow?

Response 4: We were so sorry for the mistake. We have deleted the “test-day” in the manuscript. In this study, we used the phenotypic data of 305-day milk yield, fat yield, fat percentage, protein yield, and protein percentage rather than the test-day records, which were estimated based on at least 6-10 test-day records per cow.

 Point 5: Line 72 and 73: you are referring to two lactations. Meaning that cows of first and second lactation were selected? How many in first and how many in second lactation? Clarify.

Response 5: Thanks for your comment. In this study, we selected 1,067 cows for association analysis (please see page 3, lines 69-76). There were 1,067 cows in the first lactation (327 cows merely completed their milking of first lactation), and 740 cows in the second lactation. We have added the number of cows into the manuscript (please see page 3, lines 77-78).

 Point 6: Line 94: full name of "LD" acronym is needed.

Response 6: As suggested, we have added the full name of “LD” (please see page 4, line 98).

 Point 7: Line 98: is there a specific reason why you did not inserted days in milk and parity of animals in the model? I think these two fixed effects are very important.

Response 7: Thanks for your comment. In this study, we used the 305-day milk yield, fat yield, fat percentage, protein yield, and protein percentage for association analysis rather than the test-day records. In addition, we separately analyzed the associations in first and second lactations. Therefore, we did not insert them as the fixed effects.

 Point 8: Line 99 to line 103: specify the number of levels for each fixed effect inserted in the model.

Response 8: Thanks for your suggestion. We have added the numbers of levels for each fixed effects (please see page 4, lines 104-105).

 Point 9: Line 145: To me, it is not clear what you mean for associations. Further than the significance of the association (P value), you should state and discuss whenever the association is positive or negative for each one of the traits.

Response 9: Thanks for your comment. We have revised the “Associations” to “Association analysis” (please see page 8, line 148). In addition, we set P < 0.05 as the threshold for the significance of the association. Also, we have added some discussion about the positive associations for each traits in the Discussion part (please see page 16, lines 228-233).

 Point 10: Line147: Not clear what you mean for "two, four, four, and one SNPs". Please clarify here and in the rest of the manuscript.

Response 10: As suggested, we have added the details of the SNPs into the manuscript (please see page 8, lines 150-157).

 Point 11: Line 191: Expression of LPIN1 is missing in the graph. Unit of measure of y-axis is missing.

Response 11: Thanks for your comment. We have added the missing part into the Figure 2.
Point 12: Line 210: the discussion of results sounds more like an introduction. I suggest to discuss more your results in comparison with literature instead of being so general.

Response 12: Thanks for your comment. We have rephrased the Introduction and Discussion parts (please see page 3, lines 51-61, and page 16, line 228-246).

 Point 13: Line 211 to line 213: I don t think that your work is the first on the topic (See Pegolo et al. for example).

Response 13: Thanks for your comment. We have deleted such descriptions throughout the paper.

 Point 14: Line 223: full name for "FA" acronym is needed.

Response 14: Thanks for your comment. We have changed the “FA” to “fatty acid” (please see page 3, line 59).

Round  2

Reviewer 3 Report

Authors have reviewed extensively the manuscript.

I think the manuscript is now adequate for publication.